# Evaluation Methodology for Object Detection and Tracking in Bounding Box Based Perception Modules

Paweł Kowalczyk [1,2,*], Jacek Izydorczyk [2] and Marcin Szelest [1]

1   Aptiv Services Poland S.A., ul. Podgórki Tynieckie 2, 30-399 Cracow, Poland; marcin.szelest@aptiv.com
2   Faculty of Automatic Control, Electronics and Computer Science, Silesian Univeristy of Technology, Akademicka 2A, 44-100 Gliwice, Poland; jacek.izydorczyk@ieee.org
*   Correspondence: pawel.2.kowalczyk@aptiv.com

**Abstract:** The aim of this work is to formulate a new metric to be used in the automotive industry for the evaluation process of software used to detect vehicles on video data. To achieve this goal, we have formulated a new concept for measuring the degree of matching between rectangles for industrial use. We propose new measure based on three sub-measures focused on the area of the rectangle, its shape, and distance. These sub-measures are merged into a General similarity measure to avoid problems with poor adaptability of the Jaccard index to practical issues of recognition. Additionally, we create method of calculation of detection quality in the sequence of video frames that summarizes the local quality and adds information about possible late detection. Experiments with real and artificial data have confirmed that we have created flexible tools that can reduce time needed to evaluate detection software efficiently, and provide more detailed information about the quality of detection than the Jaccard index. Their use can significantly speed up data analysis and capture the weaknesses and limitations of the detection system under consideration. Our detection quality assessment method can be of interest to all engineers involved in machine recognition of video data.

**Keywords:** quality metrics; image detection; bounding box; Jaccard index

## 1. Introduction

### 1.1. Perception Module

Perception modules use raw data streams obtained from sensors mounted on car like camera, radar or lidar devices to recognize and interpret the surroundings. Raw data collected by sensors must be properly interpreted and processed to be understood by computer. This type of analysis is carried out by algorithms supported mainly by many trained neural networks (detectors). Most perception modules based on computer vision systems use bounding boxes to mark the recognized objects in each separate frame of the video stream. Bounding boxes studied in this work are rectangles with sides aligned in parallel to the sides of the frame, and thus they can be stored as four coordinates of their opposite corners. Each perception module in a vehicle is specialized in a particular task. Bounding boxes are used to mark objects such as pedestrians, other vehicles, their lights (separately), road signs, speed bumps and traffic light signalization in streams of video data [1–6]. Based on this description, the vehicle steering system can make decisions in accordance with pre-programmed protocols and logic called ADAS (Advanced Driver Assistance Systems). It is thus fundamentally important that this description is adequate, detailed and reliable, to secure the basis on which the decisions are made.

### 1.2. Testing and Verification of Car Perception

In the process of developing perception modules, it is important to check the quality of their interpretation of data from sensors. Such verification must be carried out with high regularity if essential changes are introduced into their algorithms. Data collected by the

sensors is saved during the original data collection process which involves fleet of testing cars with mounted sensors on board and logging machines to save raw data (i.e., video from camera mounted in front of vehicle). When such data is returned to the laboratory, it can be reused in the resimulation process, which is the coordinated reconstruction of the time-ordered stream of information from sensors. Subsequently, this information is implanted into the input of the perception module version that is being tested at the moment. In this way, perception model results for given scenes are obtained. Those results consist of bounding boxes describing different elements of surrounding recorded on frames coming from the camera. In order to train detectors and verify its effects it is necessary to first describe exactly what in collected by the sensors data should be found and interpreted by the perception modules. Our reference system towards which we can compare the results that come from the detectors is called ground truth (GT). To create this reference, raw video data needs to be labeled manually which means creation of description of the expected results from the perception modules, frame by frame. This is handled by the staff of appropriately trained people who manually analyze the collected sensor data and label it. It is done according to predefined principles, this work is laborious and time-consuming. Based on these additional data, it is finally possible to calculate the quality of the results obtained from the perception module, which in a broad context enables their development and evaluation of effectiveness in real conditions. In order to reduce the time and hardware complexity of such analysis high automation is required. Well designed algorithm provides reproducibility of the evaluation results. Research on the development of the perception of smart vehicle are related to human safety and are designed to minimize the amount and damage caused by road accidents. It is therefore necessary to create methodology that will reliably and objectively assess the quality of prototypes and enable quick and effective problem localization [7]. The requirements such as, for example, SOTIF [8] (Safety Of The Intended Functionality) are created so that all car manufacturers, researchers and lawmakers could use a universal set of requirements, recommendations and good practices. This document underlines the need to confirm the effectiveness of ADAS in different situations for all functionalities those systems provides.

### 1.3. Evaluation Methodology

Huge amount of work and resources devoted to acquiring, storing and preparing them motivates the creation of evaluation methodology that make the best use of them. There is a need to ensure efficiency of development, testing and validation process of perception modules, to evaluate how well the system output depicts the stream of ground truth. This means that task involves a methodology for the comparison of two rectangles as well as sequences of them that will provide specific information relevant to context of module specialization. Amounts of data that needs to be analyzed introduces the need for full automation and repeatability of such a process. This methodology has to be clearly decisive therefore, it is important to design approaches that will allow the synthesis of detailed conclusions to use the potential of collected data. The purpose of this article is to describe a novel evaluation methodology for perception modules used in automotive vehicles. Designed solutions should aid engineers to estimate quality of detectors working on video data that comes from camera mounted at the front of the moving vehicle. It is well suited to compare GT and detectors output as a form of bounding boxes. It provides tools to assess local quality in separate pictures and summarize results in sequence of frames (tracking quality) while understanding the need of quick response in traffic conditions on the road. Presented methodology can serve as precise definition of correct recognition that highlights the fact that various types of objects, although all described by rectangles require their own special approach to evaluation. It is achieved by focusing the measures on different aspects of rectangles which allows to filter specific information about the comparison and assign appropriate meaning to it for the whole analysis. Quality measure can be successfully used as a base for matching algorithm but beside the definition of true positive it should be treated as a metric that is directly interpreted and passed to higher levels of evaluation

chain—bounding box sequence analysis. To help with interpretation of results we proposed ways to visualize output of this analysis. Both for quality of GT representation, as well as alerts of false positives which are natural consequence of matching algorithm. Methodology was presented separately for the following object classes examples: pedestrians, moving vehicles, traffic lights and signs. To achieve target of adaptation to different classes of objects and different applications in the evaluation process (matching, quality summary, combining the sequence of false positive results) methodology has to be parameters reliant. The calibration process—meaning of all parameters and their influence on final results is described in work.

*1.4. Organization of the Paper*

Next Section 2 is a review of known measures for evaluation of visual detection and tracking and possible tasks that use similar approaches. It also contains basic definitions connected to evaluation of perception modules with rectangular regions of detection. In our work, we would like to propose a new specialized tools to describe the similarity of rectangular pixel sets. Proposed tools are local measure of rectangular similarity (Section 3) and similarity of rectangle sequences (Section 4). The sequence similarity is based on an algorithm that summarizes the performance for objects regarding all their appearance on the timeline (event) during a road scenario. Next Section 5 presents application of local measures and underline problems characteristic from the perspective of automotive needs regarding interpretation of environment. It also show how awkward and misleading use of Jaccard Index in those situation can be and compare it with the defined novel solution that deals with mentioned challenges in that field. Finally, we would like to compare our measure with the Jaccard Index and different functions described in [9–11] regarding their properties (Section 6.1). Furthermore we provide results of a statistical comparison between Jaccard index and our new method. We believe that the designed massive Monte Carlo experiment (Section 6.2) shows a difference between them in a unified way that will help understand that this tool can be useful in general evaluation tasks as well and provide additional value.

**2. Related Work**

In a search for methods to measure quality for perception modules it is important to review state of the art connected to tasks revolving around the development of detectors and trackers. The most common measure of similarity used in the development of neural networks working on video data is called the *Jaccard index* (Intersection over Union—IoU), and is an example of a similarity measure. It is function $\mathcal{J} : \mathcal{P}(\mathbb{R}^n) \times \mathcal{P}(\mathbb{R}^n) \to [0,1]$, $n \in \mathbb{N}$ defined by:

$$\mathcal{J}(A,B) = \frac{|A \cap B|}{|A \cup B|}. \tag{1}$$

It is used as a tool to evaluate video object segmentation [12–14]. It work well for this task because it is a general tool to compare two irregular objects, although, even for this problem, there exist better solutions [15]. Calculations are done by dividing the number of correctly covered pixels by the sum of the pixels involved in the comparison. The problem of evaluation for bounding boxes is a specific example of the evaluation problem for segmentation and should be treated separately. Tools based on the Jaccard index formally work but they do not make use of the information that compared objects always belong to the same class with well-known basic geometrical properties. A similar criticism is found in [16], where we can also find a similar approach for constructing new measures from rectangle properties. The basic advantage of the proposed new solution is the fact that it combines a holistic approach with the possibility of decomposing information into well-isolated components. Value of such an approach is highlighted in work [17]. Most works revolve around the Jaccard Index or similar measures for the evaluation of video analysis data. Some works present comparisons of a large variety of performance measures because there is still a lack of consensus about which should be used in the experiments. For this

reason, new methodologies are constantly being developed. In recent years, there are many variations of Jaccard index (IoU) that have been proposed [18–20]. And many other approaches use it at least as a part of their evaluation methodology [21–25]. The need to construct an evaluation methodology is motivated by various problems involving detection algorithms. Authors of [26] provide examples of tracking and object detection systems for advanced driver assistance and autonomous driving applications that needs evaluation. Mean-shift tracking algorithm with spatial-color feature and parameter based Kalman filter in video target tracking [27,28] also are trackers worth describing with comparable metrics calculated with use of GT. For this task, there exists plenty of benchmark data, ground-truth, and guidelines for annotation for the unification of test and comparison of testing trackers but this goal is difficult to achieve when different evaluation metrics are used. The article [29] describes a framework for performance evaluation of face, text, vehicle detection, and tracking in video. Authors in [30] propose a new evaluation methodology which aims at a simple, easily interpretable tracker comparison. The proposed methodology accounts for the tracker equivalence by considering the statistical significance and practical differences. DAVIS Challenge, although focusing on segmentation similarly as in works, also uses evaluation methodology for the description of scene quality [31]. The use of IoU function for bounding box comparison involves the risk of missing a lot of information at the basic level of data analysis. There is a need and possibility to form a more detailed measure [16,19] that not only will refine the grade, but it will also have the properties that make the Jaccard Index widely used. Regarding the naturally connected problem of summing up the quality of object recognition over a set of video frames, as seen for example in [32], there exist different approaches than most classic one (used in DAVIS challenge). Examples are variations of the OSPA metric [33–35] which represents a possibility to divide information because it consists of three components, each separately accounting for localization, cardinality, and labeling errors. Weighted State Correlation Similarity [36], tracking difficulty [37], information theory based metric [38], shape association measure based on Minkowski distance [39], and others [40–42] are metrics for event evaluation but none of them put enough pressure on quick response which is crucial for dynamic driving situations that require immediate reactions. The Jaccard Index seems to beat the existing solutions in many ways and is widely used in various projects that require, evaluation metric. In [43], authors use it to study the effects of small errors in ground truth on the tracking algorithm results. We want to underline that, for similar cases, precision and possibility to decompose information is very important. Authors in [44] presented a combined criterion of performance evaluation by considering the veracity, real-time demand, and the implementation on hardware of the tracking algorithm, In [45], we can find an algorithm that derives fuzzy rules to merge the detected bounding boxes into a unique cluster bounding box that covers a unique object. Authors underline the need for a tool that describes relationships of a pair of boxes by their box geometrical affinity, by their motion cohesion, and their appearance similarity. Object tracking in realistic scenes is a complex problem and it remains an active area of research in computer vision [46]. Another popular use of the similarity measure is to treat it as loss function for training tracker. Work [47] shows the method for optimization of the mean intersection-over-union loss in neural networks. The loss is shown to perform better with respect to the Jaccard index measure than the traditionally used cross-entropy loss. The need for the evaluation metric is represented by both the tracker evaluation and fusion evaluation. It is why authors of [48] define an evaluation methodology that can also be applied in this field. The proposed context metrics are to support standard metrics to deal with various challenges that stand in way of correct evaluation of tracker/fusion algorithms.

## 3. Rectangular Similarity

To start our considerations about similarity, it is worthwhile to recall the mathematical definition of such concepts. We can say that for given set $\mathcal{R}$, the function $s : \mathcal{R} \times \mathcal{R} \to \{p \in \mathbb{R} : 0 \leqslant p \leqslant 1\}$ is a *similarity measure* if for any $x, y \in \mathcal{R}$, it satisfies the two conditions:

- $s(x, y) \leqslant s(x, x)$
- $s(x, y) = s(y, x)$.

  It is worthy to notice that a third condition:

- $s(x, y) = 1 \Leftrightarrow x = y$,

although is not formally required, is highly valuable. Especially in our situation, since later we will combine measures to create new ones. In practice, the usefulness of such a function is demonstrated by the fact that its value changes informatively and is as close as possible to the intuitive concept of similarity in the considered case (bounding box comparison in our case). To construct a new similarity measure for bounding box comparison, we create three sub-measures that can evaluate the similarity of two rectangles in terms of three main attributes: area, shape, and distance. Later, based on those functions, we construct a general similarity measure that will fulfill all conditions mentioned at the beginning of this section. Those sub-measures should obtain their values as independently from each other as possible. This is because any difference between two parallel rectangles can be fully described as a combination of deviations in those three fields.

### 3.1. Area Similarity

Let $\mathcal{R}$ be the set of all rectangles in $\mathbb{R}^2$. Although, in this work examples regarding video data contain only boxes parallel to sides of picture, this assumption does not restrict the applicability of described methodology to more general tasks (see Section 5.4). For two rectangles $L, R \in \mathcal{R}$, we define their *Area similarity* as:

$$A(L, R) = \frac{\min(P(L), P(R))}{\max(P(L), P(R))}, \tag{2}$$

where $P$ denotes the value of the area of a given rectangle. This means that we obtain 100% similarity only when both rectangles have exactly the same area. The shape and position of rectangles have no influence on the value of this function.

### 3.2. Shape Similarity

In our work, we consider rectangles with a predefined width and height. This allows us to set a convention about the angle between the diagonal and the side of the rectangle that is considered as the width. This angle defines the shape of the rectangle. For two rectangles $L, R \in \mathcal{R}$, we define their *Shape similarity* as

$$S(L, R) = \cos^p(\alpha - \beta), \tag{3}$$

where $\alpha$ and $\beta$ are the values of the angles defining the shape of the rectangles $L$ and $R$, respectively. The parameter $p \in \mathbb{R}^+$ serves to reduce the value of the function depending on the need to penalize small deviations. To achieve goals described in Section 3.5 different values were tested and parameter $p = 17$ proved sufficient effectiveness to both stay informative in terms of shape evaluation and the low overall importance of low shape discrepancies in the final result. Similarity equal to 100% in this measure will be achieved only when the mentioned diagonals are parallel and any deviation from this situation will cause a lower cosine value and a decrease in the similarity rate. Position and size of rectangles have no influence on the values of this function.

### 3.3. Distance Similarity

Let $(\mathbb{R}^2, d)$ be a metric space. The function $s()$ which maps real numbers $\mathbb{R}$ into $[0, 1]$ interval can constitute a *similarity measure based on metric d* if, for any $x, y, z \in \mathbb{R}^2$,

$$d(x, y) \geqslant d(x, z) \Rightarrow s(d(x, y)) \leqslant s(d(x, z)). \tag{4}$$

The additional property that is appropriate for the similarity measure based on distance is that

$$d(x,y) = 0 \Leftrightarrow s(d(x,y)) = 1. \tag{5}$$

Let $m : \mathcal{R} \to \mathbb{R}^2$ be a function that returns its centroid for any given rectangle. $d$ is Euclidean distance. For two rectangles $L, R \in \mathcal{R}$, we could define their *distance similarity* (*position* similarity) as the value of the function

$$D(L, R) = \exp\left(-\gamma d(m(L), m(R))^\delta\right). \tag{6}$$

It is easy to check that for positive constant parameters $\gamma$ and $\delta$, the above mentioned conditions (4) and (5) are fulfilled. Focusing this measure on the detection center defined this way ensures independence from the shape and area of rectangles. This formal independence will be later intentionally reduced to local independence on certain subsets of $\mathcal{R} \times \mathcal{R}$ in order to achieve informative values that will appeal to intuitive approach to the concept of distance between objects. Objects far away from the camera should be assessed analogously to objects close to the camera, despite the much shorter Euclidean distance between the pixels that define their center of gravity. Therefore, we propose scaling measure of distance between centers on the basis of the size of the compared objects. This means that the measure of distance is indirectly related to the size of the rectangles to be compared.

Simultaneously, parameters $\gamma$ and $\delta$ will be used as control values of the distance similarity function in a good fit and in a bad fit domain, with a swift transition between them. To define $\gamma, \delta$, let us assume that we expect our function to produce value $s_1$ when the scaled distance between centers of rectangle $d(L, R)$ is equal to $p_1$ and produce value $s_2$ when the distance is equal to $p_2$. Let us further enumerate the following basic assumptions about those parameters

$$p_1, p_2 \in \mathbb{R}^+ \quad \text{and} \quad 0 < p_2 < p_1, \tag{7a}$$

$$s_1, s_2 \in (0, 1) \quad \text{and} \quad 0 < s_1 < s_2. \tag{7b}$$

To find explicit formulas for $\gamma, \delta$, we solve the following linear system of equations

$$\begin{cases} s_1 = e^{-\gamma p_1(L,R)^\delta}, \\ s_2 = e^{-\gamma p_2(L,R)^\delta}, \end{cases} \tag{8}$$

to obtain

$$\begin{cases} \delta(L, R) = \ln\dfrac{\ln s_1}{\ln s_2} \Big/ \ln \dfrac{p_1(L, R)}{p_2(L, R)}, \\ \gamma(L, R) = \dfrac{-\ln s_1}{p_1(L, R)^{\delta(L,R)}}. \end{cases} \tag{9}$$

These equations allows us to compute parameters $\delta$ and $\gamma$ to uniformly codify values of the similarity measure versus the scaled distance between two centers of rectangles (Figure 1). Coding, based on the values $s_1, s_2, p_1, p_2$, allows us to define which fitting we consider as acceptable and which as critically bad. The definition of $p_1, p_2$ can be tailored to our needs, so long as it fulfills the basic assumptions (7a) and (7b). For testing purposes below we established $s_1 = 10\%$ and $s_2 = 90\%$.

The final step is to calibrate the position similarity measure that is, determining the relationship between the parameters $p_1, p_2$, and the sizes of the compared rectangles as mentioned in the beginning of this section. The function should reflect the size of rectangles $L, R$ in the measurement. The best distance within the rectangle boundaries to describe overall size of rectangle is the length of its diagonal. Therefore, we proposed definition of $p_1$ and $p_2$ as a weighted sum of diagonal lengths of rectangles under consideration $L, R$, for example:

$$p_1(L,R) = \frac{4}{10} diag(L) + \frac{2}{10} diag(R), \tag{10}$$

$$p_2(L,R) = \frac{2}{10} diag(L) + \frac{1}{10} diag(R). \tag{11}$$

where $diag : \mathcal{R} \to \mathbb{R}$ is the function that for any given rectangle returns length of its diagonal. We have chosen the weight coefficients to obtain a reasonable maximum allowable displacement between rectangles for pedestrian cause, scale parameter should be chosen based on practical needs. For example in case of mobile vehicle detection scale function should have different constant parameters and involve width of bounding boxes instead of diagonal length. The functions $p_1(L,R)$ and $p_2(L,R)$ should take into account the specificity of a given module perception.

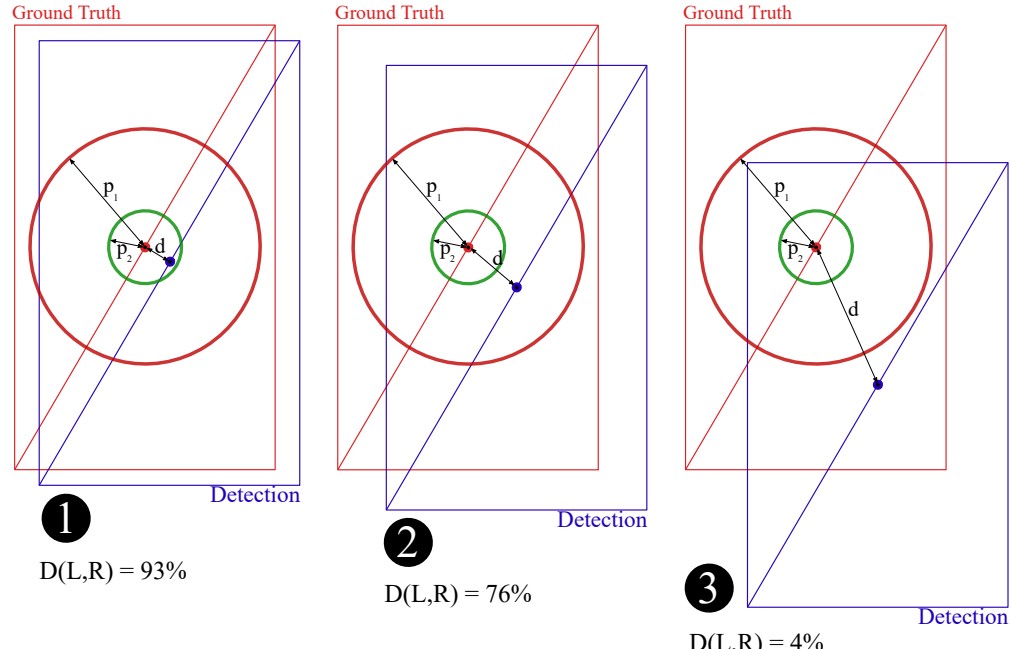

**Figure 1.** Distance similarity represents the difference in the position of rectangles rooted in the context of sizes of compared boxes.

The only awkwardness of the definition of the distance measure function is that it is not invariant on rectangles exchange:

$$\exists_{L,R \in \mathcal{R}} \quad D(L,R) \neq D(R,L). \tag{12}$$

In the case of twins, Adam and Eve, it is unusual to say that Adam is more similar to Eve than Eve to Adam. Obviously, we can overcome the effect by choosing equal values of weights in Equations (10) and (11). We have not done it because we want to underline that bounding boxes taken from ground truth (GT) are more trustworthy than bounding boxes detected by sensors. Therefore, they should have grater (perhaps even exclusive in some cases) influence on the evaluation function than other ones. First argument of the similarity function must be taken from GT. It is our motivation to keep an unusual, unsymmetrical distance similarity function but with assumption that we feed this function with arguments in the established order. Thanks to this we can still consider this function as symmetrical in practical sense.

For this definition of the similarity measure, as we will be able to see, we have real control over the independent influence of GT and sensor results on our evaluation method for good and bad matches. All of the in-between observations are evaluated based on

one curve depicted in Figure 2. The function scales individually for each comparison. It is worthwhile to notice that to achieve presented convexity for low distance values, the calibration parameters should fulfill the additional assumption $\delta > 1$. In practice, as long as $s_1$ and $s_2$ are distant enough from each other (which is natural to achieve informative values), we have a relatively free choice of parameters $p_1, p_2$.

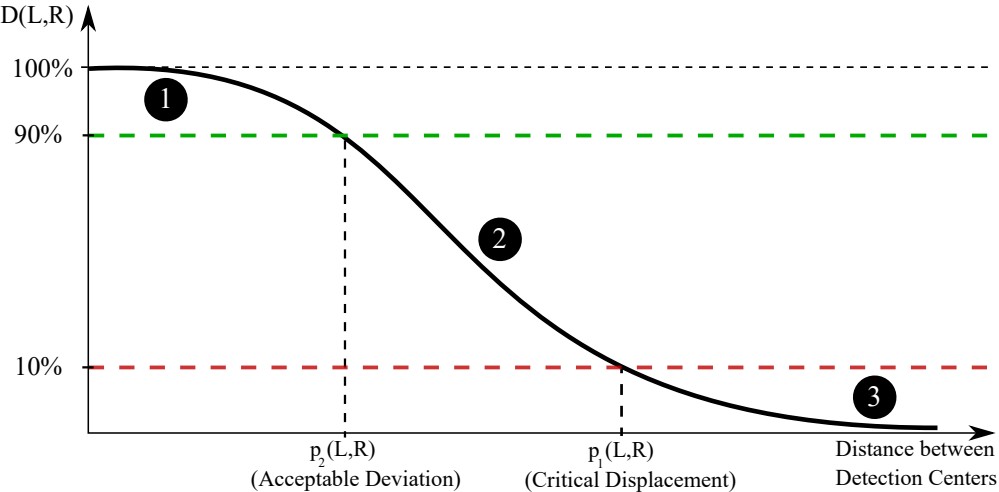

**Figure 2.** Distance Similarity curve. The argument of this function is the distance between the detection centers. The result for position quality measure is obtained after rescaling the function curve specifically for each comparison. Parameters $p_1$ and $p_2$ describe the main thresholds for evaluation of boxes properties.

### 3.4. General Similarity

Based on the three aforementioned sub-measures, we define a general measure that describes the similarity of two rectangles from $\mathcal{R}$, and fulfills the basic and additional requirements of our measure definition. For $L, R \in \mathcal{R}$, we define the *general measure of similarity* (GMOS) as

$$GMOS(L, R) = \frac{3}{\frac{w_1}{S(L, R)} + \frac{w_2}{A(L, R)} + \frac{w_3}{D(L, R)}}, \tag{13}$$

where

$$w_1, w_2, w_3 \in \mathbb{R}^+, \tag{14a}$$
$$w_1 < w_2 < w_3, \tag{14b}$$
$$w_1 + w_2 + w_3 = 3. \tag{14c}$$

This measure will achieve 100% only when two rectangles are the same in terms of all sub-measures. The harmonic mean is used because its value decreases drastically once the distance similarity is low. To achieve this with the arithmetic mean, we would have to use weights for the shape and area similarity measures which are much lower than the weight for the distance similarity measure, which would effectively make those two measures insignificant.

### 3.5. Accuracy of Association Algorithm and Selection of Hyper-Parameters

With these tools, we can write an algorithm that will associate proper results with GT boxes based on distance similarity and secure a proper fitting with other sub-measures. Based on this methodology, we can define what can be considered as the true positive in a more precise way, provide more insight about box association quality, recognize truly redundant results and mark them as false positives. In order to confirm this statement

we run two association algorithms, first demanding $GMOS(L, R) > 10 \wedge A(L, R) > 25\%$ and second demanding $IoU(L, R) > 30\%$. Those two analyzed over 100 pictures with 700 bounding boxes of detected pedestrians on the road with prepared GT. Then we analyzed manually how many connected pairs of boxes are correct true positives reported with those two approaches. Accuracy of this method for determination of true positives is equal to 98.2% in comparison to Jaccard Index which marked correctly 94.3% as true positive. Lowering threshold for Jaccard Index causes risk of accepting critically bad recognition as true positives (which is discussed deeply in Section 5), this can lead to overestimated precision for detector. Precision statistic is insensitive to information about local quality results and this is why in next section we propose alternative method of summary of the local quality for sequences of pictures.

It is important to understand that GMOS method heavily relies on hyper-parameters which allows for precise definition for true positive and proper evaluation but has drowbacks. Badly chosen parameters can provide misleading results so choosing them carefully is very important. Weights used by authors are

$$w_1 = \frac{2}{7}, \qquad w_2 = 1, \qquad w_3 = \frac{12}{7} \tag{15}$$

which shifts most of the significance from shape aspect to position. Those weights along with other parameters (like $s_1$ and $s_2$ connected to the scaling of distance similarity measure and the way final results are reported and interpreted) associated with this methodology are concluded by authors based on the expertise of engineers working in Aptiv on improvement and evaluation of perception modules and are best suited for this purpose as proved in Section 5. Specific parametrization of $p_1$ and $p_2$ for distance similarity is proposed for pedestrian evaluation case and results of artificial experiments in Section 6.2 are tied to it. Different parametrization is used for vehicle and traffic signs in section with practical applications.

## 4. Rectangle Sequence Evaluation

To evaluate the quality of a single object detection during its whole appearance in time, we summarize the results of the defined similarity measures on the span of frames in which the GT box sequence describing the object exists.

### 4.1. Weighted Average for Late Detection

Let the time series of rectangles taken from GT $\mathcal{L} = \{L_1, \ldots, L_n\}$ represent the object that appears in the span of $n$ frames. The object inside $i$-th frame is bounded by a rectangle $L_i \in \mathcal{R}$. Let $I \subset \{1, \ldots, n\}$ be a set of all frame indicators for which there is a corresponding recognition for $L_i$, $i \in I$, i.e., a rectangle $R_i \in \mathcal{R}$ that bounds the same object as $L_i$. Let $\mathcal{P} = \{R_k, \ldots, R_l\}$ be the time sequence of all rectangles that can be connected to the correct $L_i$ for $i \in I$. A measure of similarity $SGMOS$ that compares manually tagged (GT) events $\mathcal{L}$ with those detected by the system under consideration $\mathcal{P}$, can be:

$$SGMOS(\mathcal{L}, \mathcal{P}) = \frac{1}{|\mathcal{L}|} \sum_{i=1}^{|\mathcal{L}|} w_i o(i) \tag{16a}$$

where

$$o(i) = \begin{cases} G(L_i, R_i) & \text{for} \quad i \in I \\ 0 & \text{otherwise} \end{cases}, \tag{16b}$$

and $w_i$ are weights calculated taking into consideration the time needed by the system to react to the appearance of a new object.

Sensor systems usually need to see an object in several frames before the first detection. That is why we used weights to limit the influence of these frames. Weights are also used to assign penalty to the system for not detecting objects in the expected time. These two

factors lead us to define two formulas on the basis of which weights are calculated. But first let us define:

1. *Critical index* ($CI$) is the number of frames from the moment the object appeared within the range of the sensors in which detection delay can be tolerated. Simultaneously the detection appearing not later than $CI$ is crucial for safety.
2. *First detection* ($FD$) is the first frame in which recognition of the object appears and we are able to evaluate the quality of the detection by the use of the similarity a measure applied to the object and the corresponding GT object.
3. *Standard weight* (SW) is the weight that will be assigned to each observation from the first detection to the end of the time series $\mathcal{L}$.

The method of calculating the weights depends on the location of the first detection in relation to the critical index. Let us consider two possible cases. The *first case* occurs when the first detection happens before the critical index. For indices between 1 and $CI$, the weights will increase linearly from 0 for index 1 to 1 for index $CI$. In this case, the weights before $FD$ will be taken appropriately from this set. The weight of frame $FD$ and all others is equal to $SW$, even if the sensors lose their detection at some point. This way, the lack of signal does not affect the result so dramatically. This is a desirable behavior, because we can assume that after fusion with data from other sensors, certainty as to the presence of this object is already high.

$$w_i = \begin{cases} \dfrac{i-1}{(CI-1)} & \text{for} \quad 0 \leqslant i < FD \\ SW & \text{otherwise} \end{cases} \tag{17a}$$

where

$$SW = \frac{2(CI-1)|\mathcal{L}| - (FD-1)(FD-2)}{2(CI-1)(|\mathcal{L}| - FD + 1)} \tag{17b}$$

The *second case* occurs when the first detection takes place after the $CI$ frame. In this case, it is assumed that the observation's weights before $CI$ are the same as in the first case. Weights after $CI$ increase linearly from 1 for the $CI$ frame to $k \cdot SW$ for the frame $FD - 1$. The $k > 1$ constant can be selected based on how severely we want to penalize the system for late detection. The weight for $FD$-th frame and for all subsequent ones is equal to $SW$. Second case is presented in Figure 3.

$$w_i = \begin{cases} \dfrac{i-1}{(CI-1)} & \text{for} \quad 0 \leqslant i \leqslant CI \\ \dfrac{(i-CI)(k \cdot SW - 1)}{FD - CI - 1} + 1 & \text{for} \quad CI < i < FD \\ SW & \text{otherwise} \end{cases} \tag{18a}$$

where

$$SW = \frac{2|\mathcal{L}| - FD + 2}{2|\mathcal{L}| - 2FD - CI \cdot k + FD \cdot k + 2}. \tag{18b}$$

In both cases, the sum of all weights must be equal to $|\mathcal{L}|$ (i.e., the number of frames in which the object appeared). This ensures that the $SGMOS$ function is a measure of similarity and that the detection performance for any pair of long time series $\mathcal{L}, \mathcal{P}$ can be directly compared with the performance for any pair of short time series. Visualization of local quality measure and values of $SGMOS$ are on Figure 4. In addition, it is possible to dynamically redefine $CI$ depending on the traffic situation. For example, we can manipulate the position of $CI$ based on the speed of our vehicle, and the weights will still be calculated according to the described algorithm. Finally, we can modify the weights of the general measure to highlight any component of it.

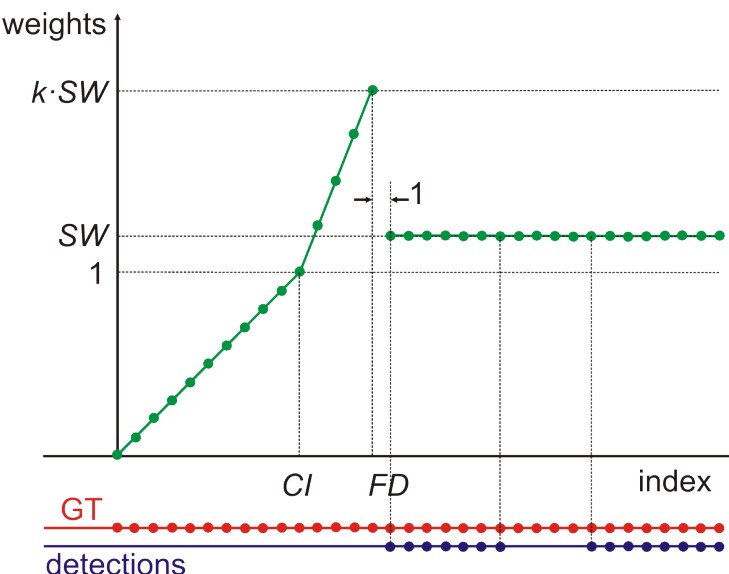

**Figure 3.** Weights calculation algorithm is designed to focus the evaluation type on the position of the first detection. This ensures that the calculation of mean will not blur the evaluation for the long appearance of objects, the results of the measure will be comparable for different events and its value will focus on the most crucial moment from the perspective of the automotive vehicle.

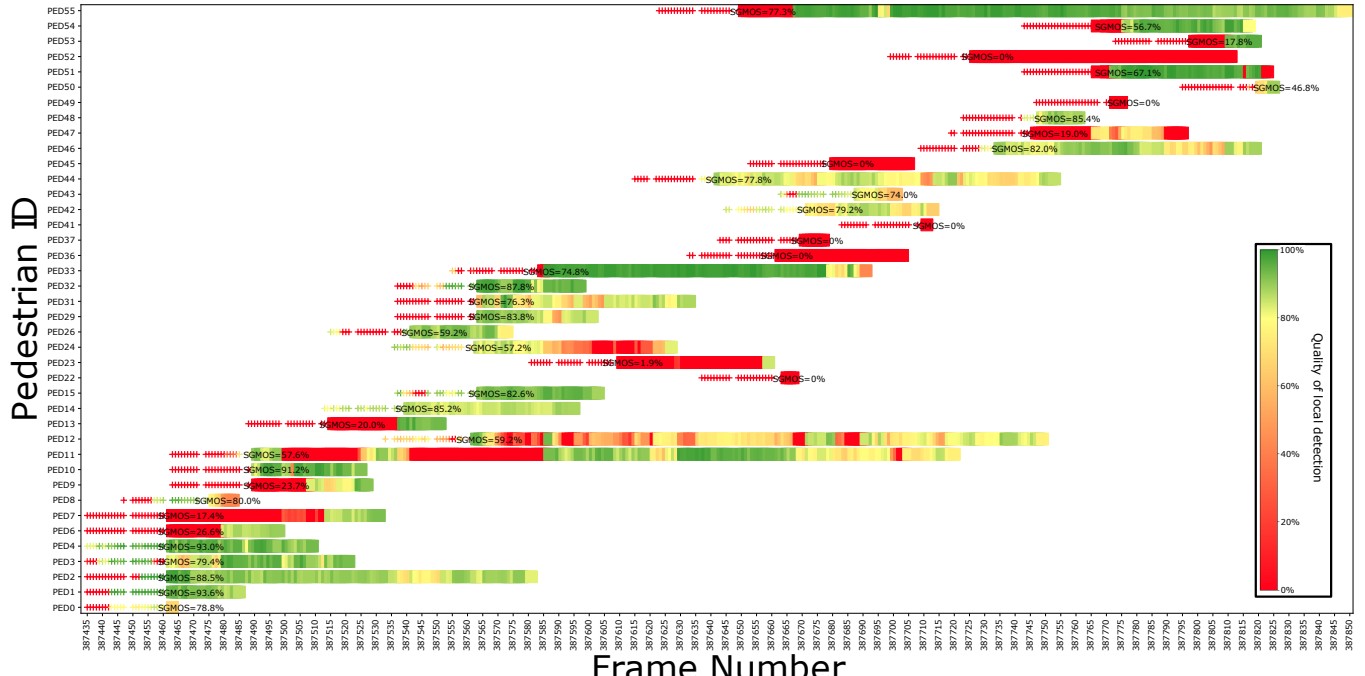

**Figure 4.** Visualization of results for the event evaluation. Each bar represents the quality for a single object during its appearance in the camera field of view. Crosses mark the state of detection before the Critical Index for a given event. Values of SGMOS for given sequence are on each bar.

## 4.2. False Positive Event

False positive results—bounding boxes connected to no objects—should not be analyzed in the context of individual frames. If false positives appear in a region only for a short time, they are considered harmless and ignored. Long-term false positives sequences should be considered as events. We can use the constructed similarity measure to match the false positive results between successive frames. Usually, it is necessary to reduce the weight of the distance similarity to connect elusive detections. Finally, if we

cannot find similar false positive results in the next frame (or several frames), we can break the event, and report its length, average width, height, and position. Additionally, we can check whether the beginning of this false positive event has its roots in the GT that has ended in the last frame before the initial false positive result appeared (similarly for results rooted at the beginning of the event). Usually, those situations are not problematic either. With this kind of approach, we can perform an automated initial analysis, summarize the results, and help with prioritization of cases (Figure 5) which will speed up any following manual analysis substantially.

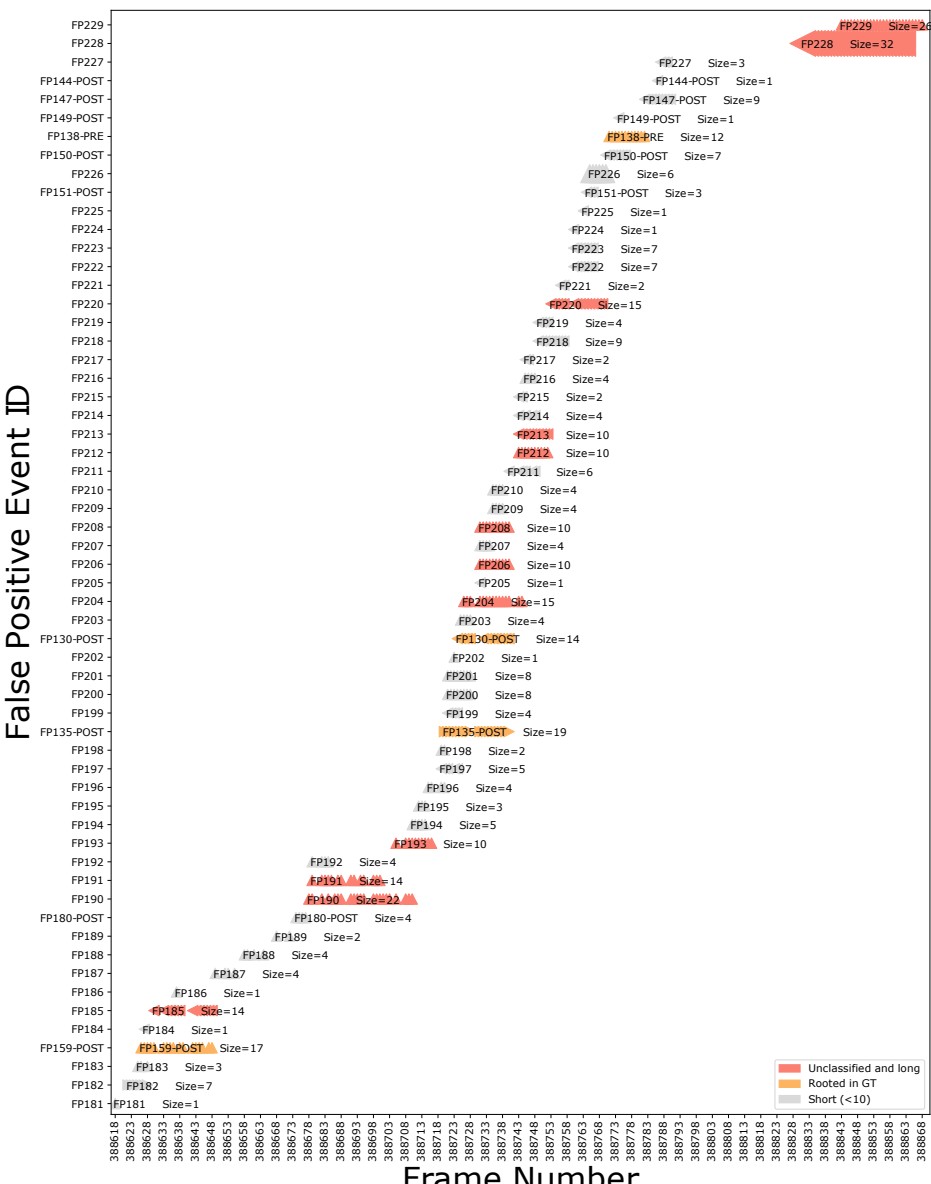

**Figure 5.** Visualization of results for false positive event analysis. Orange color marks false positives rooted in GT. ID for those events contains ID of the original GT event and information whether the false positive is rooted in the beginning (PRE) or at the end of event (POST). Gray events are not stable and long enough to be considered significant. Red events are unclassified and long, thus have the priority in further analysis.

### 4.3. Automotive Aspect of Tracking Analysis

Data used in our experiments contains videos with 30FPS which means that the amount of individual video frames that our camera captures per second is equal to 30. Part

of this has to be accepted as the latency for vision algorithms implemented in the car and is reflected by parameter CI in our algorithm. But why do we need to punish the value of similarity measure so severely when the first detection exceeds CI? Motivation for this approach from the point of the automotive and the active safety system evaluation is that the appearance of the new object is usually corresponding to the significant change of the situation on the road. It is connected to the most decisive action for control protocols in the car and securing the fact that the virtual environment is precisely described in that moment is crucial [49,50]. The fact that, while driving a car, the reaction time reduced by fractions of a second may cause drastic changes in the effects of the maneuver carried out, should not require additional explanations. Test drives are conducted safely in a way that even in situation when the detecting algorithm, data are still collected. This test drive recording is later simulated repeatedly with improved detection algorithms. Now assume that we use classic average value for all observations from video and let us consider the situation presented in Figure 6.

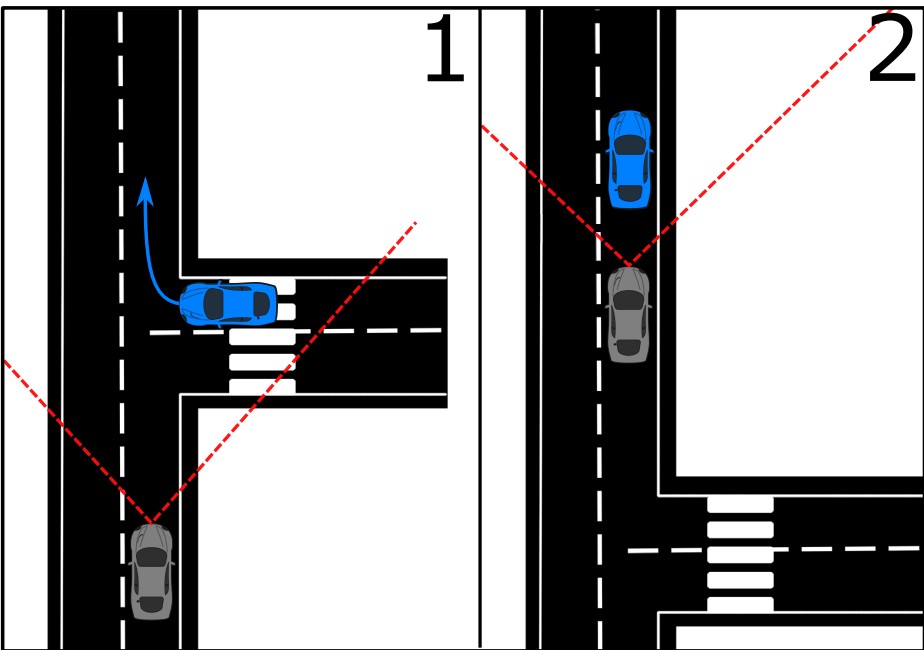

**Figure 6.** An example of a situation on the road when the late first detection can correspond to omitting a crucial event during the whole appearance of the object. 1—The initial maneuver of joining the traffic of an object. 2—Following the object. Our tracking evaluation approach is constructed to indicate this situation and emphasize it correctly.

On the first half, we see a car that enters the field of view of host vehicle by joining traffic at an intersection. Let us assume that the algorithm misses it during a major part or the whole maneuver which would normally require the host car to immediately adjust the speed or change lanes. During the simulation, we will in that moment acquire zeroes from similarity measures for a few seconds. After that, the road situation is stabilizing, let us assume that the vision algorithm catches the correct detection and keeps it from that point. The object moves in front of us for several minutes correctly recognized as seen on the second half of Figure 6. The average value of similarity measures of any kind collected for this object will be high because of the disproportion of correct recognition from the rest of the scene which reduces the impact of zeros from the initial contact. This behavior is unacceptable. Measure proposed by us addresses this problem appropriately. To experimentally present this issue, we have designed an artificial situation corresponding to this. Let us assume that we have a measure of similarity that can be used to describe quality in frames. We will compare performance analysis of tracking for this measure using standard mean approach, OSPA metric, and our method (referred to as CIvsFP). Let

us assume that the maneuver of joining traffic depicted on the left side of Figure 6 lasts 5 s. We use a camera with 30FPS which means that we will get 150 results of measure for all frames in that period. Let us assume that our algorithm missed the object for initial 2.5 s (75 frames with measure value equal to 0%). Then it works perfectly (75 frames with measure 100%). Now we will be adding observations containing our car following the one that joined the traffic (the right side of Figure 6). We will add 5 more seconds in each step until we reach 1 min and for each step we will measure the quality of the whole event with both methodologies (assuming $CI = 0.1$ s). Additionally, we will perform a supplementary analysis for this event lasting 3 min (Because it is the maximum length of a testing scenario). Results are presented on Figure 7. As we could expect the the classical mean value will initially summarize quality of this recognition to 50% and quickly rise to values over 90% suggesting in the final results that we are dealing with an almost perfect detection. In fact, missed detection, that causes major hazard, is overwhelmed by correct detection from a more trivial period of following an object. On the opposite, we can observe how the built-in protection in our CIvsFD-method prevents us from obtaining misleading results in this kind of situation. The last, "outlier" comes from the scene that lasts 3 min and is the limit of the length of the scenario. It is worth to underline that this approach is naturally focused on automotive application.

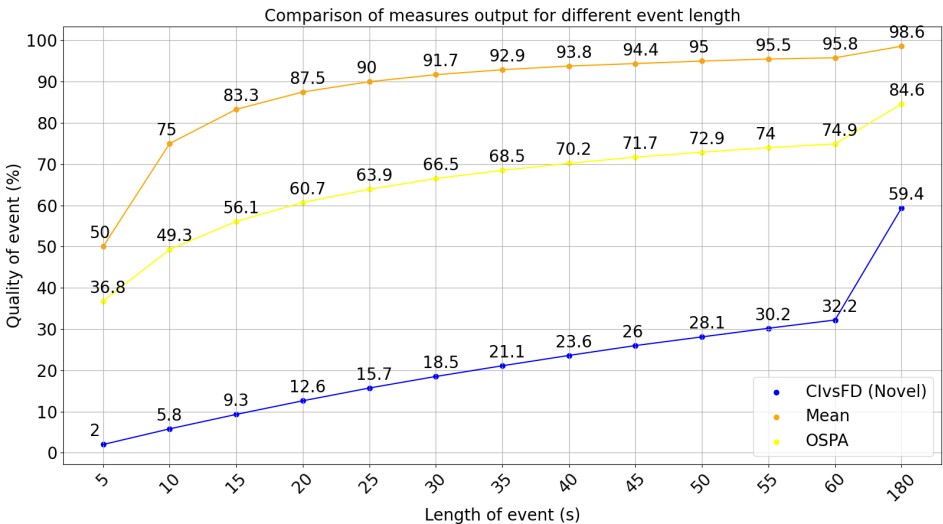

**Figure 7.** Comparison of tracking analysis measured with our approach and standard mean for situation with badly recognized initial maneuver (2.5 s).

## 5. Results of Practical Application

In this section we would like to present examples of local measures results with practical calibration suited for different functionalities. Additionally we would like to compare this results with Jaccard Index values and based on it underline the advantages of the new approach.

### 5.1. Pedestrian Detection

Below we consider pedestrian recognition from Figure 8. The need to detect such objects is not only related to the automotive field, but it is also part of intelligent monitoring connected to public safety [51]. Values of the compared measures for this situation are stored in Table 1. The proposed rectangular measure is invariant from the distance of objects from vehicle which means that position displacement is translated by the size of the object. Thanks to the proper calibration and possibility to see values of general measure in the context of sub-measures, we are able to automatize the highly precise quality evaluation. For this function example, the box that describes the pedestrian with the number 221 (second from the left) has the wrong position, shape, and area. Both general measure and Jaccard index return a similarly poor evaluation for this recognition, but the low value of

the Jaccard index is only due to the excessive size of the automatically detected bounding (blue) box. The situation of pedestrian number 233 (the first one from the right at the pedestrian crossing) is similar, but the position of the blue box is correct. The Jaccard index evaluates this recognition even worse than the previous one. The general measure still considers this recognition to be bad, but the numerical value is much higher because it is actually a better recognition than the previous one. Pedestrian with number 239 (third from the left) is described by a box with the proper size and shape but its position is shifted. Despite the fact that the results differ by only a few pixels, the shift should be considered significant due to the long distance between objects and the vehicle. The biggest problem with using the Jaccard index to the pedestrians is that the effect of the missed box's width is multiplied by its height. Simply ignoring the height is not enough, as it can lead to the loss of position information. The area similarity suffers from the same problem (in the situation when one box is fully contained in another, the value of this measure is exactly the same as the value of the Jaccard index—case 233 is an example). However, in the context of the other proposed similarity measures, it can be assessed whether the poor area similarity is indeed harmful. Let us consider the key case from the point of view of car control protocols—pedestrians 231 and 132. They both cross the street in front of the car. Pedestrian 231 is on the car's stop line and 132 is on the pedestrian crossing. The width of pedestrian 231 is perfectly described by the blue bounding box, but the height is critically wrong (see the bottom part of the box), which can lead to a misleading and dangerous conclusion that this person is crossing the street correctly. Box 132 is slightly misplaced in relation to GT for all borders, but can still be considered an almost perfect recognition. The Jaccard index evaluates both cases almost identically—80.4 percent for poor detection and 79.1 percent for perfect detection. The general measure of similarity describes poor detection with 39.5 percent and excellent detection with 94.5 percent. A similar comparison can be made for pedestrians 237 and 238, but they appear in the background. The general measure of similarity provides values that better describe the detection process in terms of output quality than the Jaccard index. Furthermore, the example confirms that the proposed general measure of similarity is scale invariant, which seems to be one of the most desirable properties for this type of quality measure.

**Table 1.** The rectangular similarity and the Jaccard index for pedestrians detected in Figure 8.

| PED | Jaccard | General | Distance | Area | Shape |
|------|---------|---------|----------|-------|-------|
| 221 | 43.6% | 41.3% | 37.0% | 43.6% | 85.3% |
| 233 | 39.0% | 63.3% | 99.0% | 39.0% | 64.4% |
| 239 | 36.7% | 47.4% | 34.1% | 98.6% | 99.9% |
| 231 | 80.4% | 39.5% | 28.3% | 80.4% | 97.8% |
| 132 | 79.1% | 94.5% | 99.8% | 85.8% | 97.7% |
| 237 | 70.6% | 35.9% | 25.2% | 80.7% | 97.0% |
| 238 | 56.1% | 86.4% | 78.4% | 100% | 100% |
| 232 | 69.4% | 86.8% | 99.7% | 69.4% | 95.9% |
| 342 | 80.4% | 97.4% | 96.2% | 98.9% | 98.8% |
| 341 | 70.4% | 97.8% | 96.3% | 100% | 100% |
| 340 | 0% | 0% | 0% | 0% | 0% |

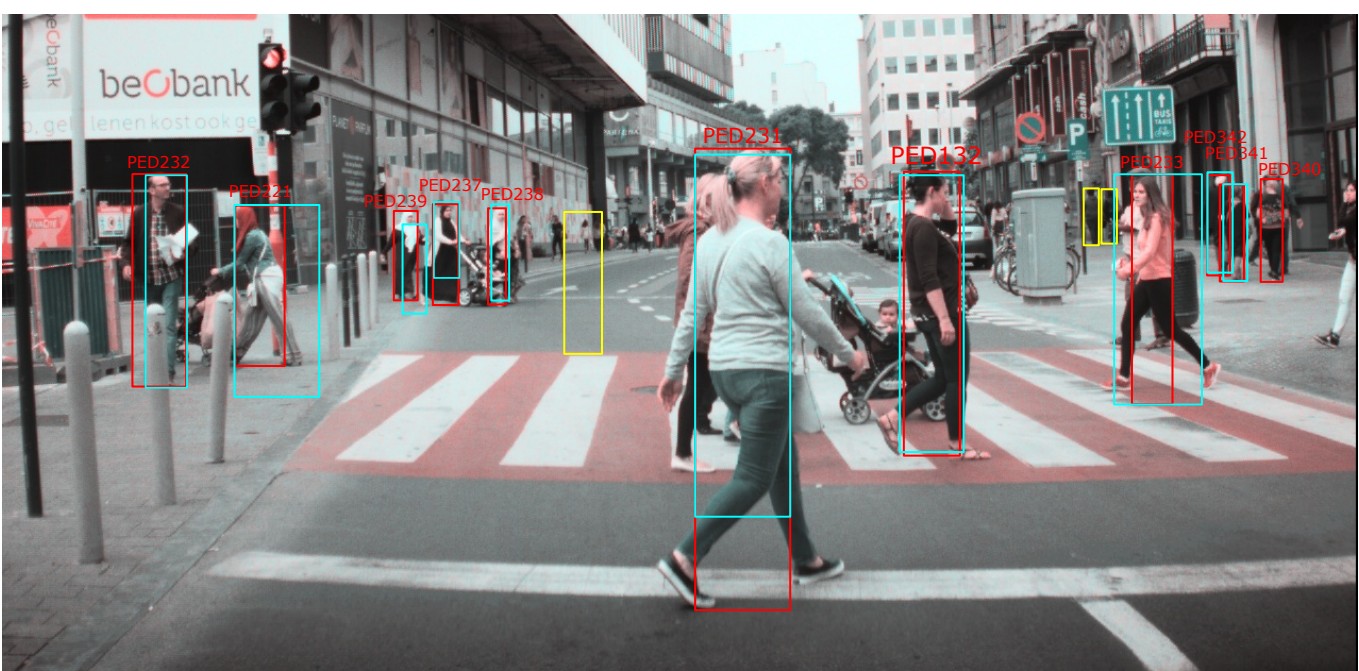

**Figure 8.** Results of Pedestrian Detection. Red rectangles represent ground truth for objects with a given ID. Blue rectangles represent the detection algorithm output that has equivalents in GT. Yellow rectangles are false positives results. In the pedestrian case, where boxes are typically strongly oblong, deviations on the base should be significantly influential. However, the Jaccard Index puts more importance on missed pixels in horizontal direction than in vertical. The distance measure (and the general measure too) behaves in the reverse sense. See Table 1.

*5.2. Vehicle Detection*

Another problem, that may arise when assessing the fit of the bounding box, is that sometimes we can obtain results that are fundamentally different from the ground truth, but we would like to consider them as correct, just like those that fit GT perfectly. The general similarity function can be very helpful in this situation, because we can change the definition of the detection center according to the region that is key in a given situation. The best example of this problem can be found in detecting tall vehicles such as trucks. Sometimes at night, test sensors detect only the lower part of the truck that contains the rear lights, instead of detecting the whole truck—see Figure 9. As this is enough to make the car driving system to behave properly, we would like to have a tool that evaluates the detection positively, knowing that the fit is not perfect. To automate the handling of this situation, we add the possibility of lowering the compared detection centers from their default positions to our distance assessment. First, we recognize if the vertical position of the center of the box is higher than the vertical position of the center of the corresponding box in GT. If this is true, no correction is required. Otherwise, we move the detection center of both boxes, i.e., for GT and for the result, depending on the ratio of the height $H$ and width $W$ of the box. The higher the ratio, the more significant the movement of the detection center from center of the mass. The offset values can be based on the inverse-logit curve to maintain the original detection center position for objects that are not tall. The offset limit is a fifth of the height.

$$y_{\text{new}} = y_{\text{old}} - \frac{H}{5 \cdot (1 + \exp(-H/W))}, \tag{19}$$

where $y_{\text{old}}$ and $y_{\text{new}}$ are correspondingly ordinate value of the detection center before and after correction. These corrections are applied independently for the GT and the result. This means that when the height difference between the boxes is small, both detection centers will be shifted in a similar way, and the value of the similarity function will not change significantly compared to the original state. When the system recognizes only the

lower part of the GT, the detection center shift will not be significant, while the detection center of the GT will be in its lower part. This will cause the distance similarity function to have a reasonable value. The information that this match is not perfect will be visible after checking the value of the area similarity and the value of the shape similarity. Examples can be seen in Figure 9 and Table 2. By shifting the center of each object, we correct the value of distance similarity and thus ensure that we do not classify the result as a false positive and unrecognized object, but as intended as a single, correct object.

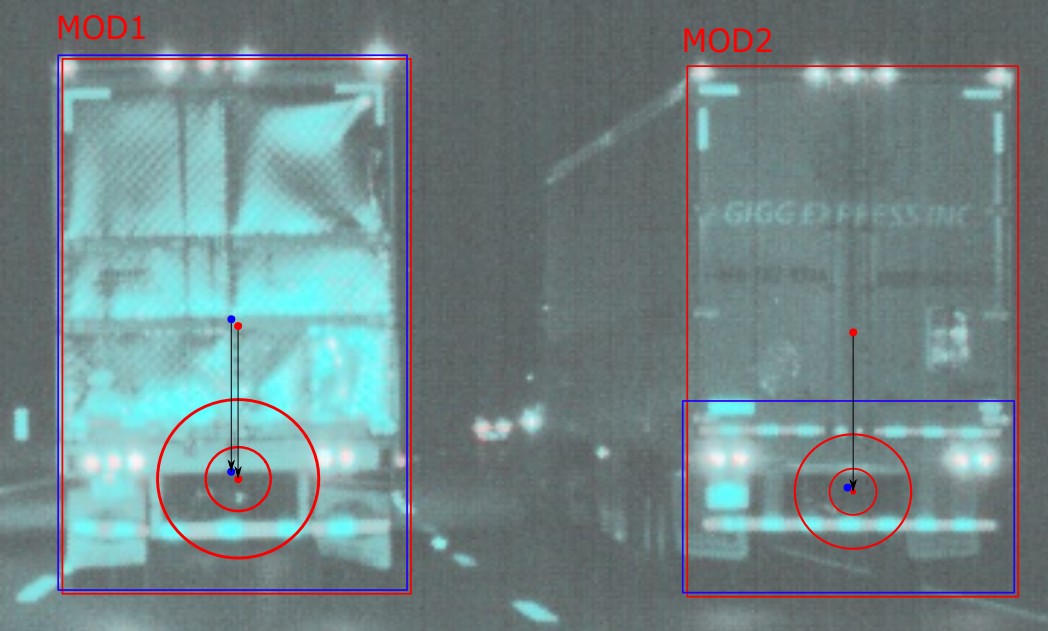

**Figure 9.** Results of Vehicle Detection. To properly evaluate detection of the most crucial segment for this function, the algorithm considers different detection centers that relocate depending on the type of the object and size of the GT bounding box data.

**Table 2.** The rectangular similarity and the Jaccard index for moving objects detected in Figure 9.

| MOD | Jaccard | General | Distance | Area | Shape |
|-----|---------|---------|----------|------|-------|
| 1 | 94.4% | 99.5% | 99.4% | 99.5% | 100% |
| 2 | 32.7% | 36.6% | 98.5% | 33.0% | 9.1% |

### 5.3. Traffic Light Recognition

A similar problem occurs in the recognition of traffic lights. Originally, we required the system to include a housing for each light, regardless of which one is on. However, at night, we can expect the sensors to recognize only the glowing parts. In this case, we would like to assess the detected boxes as acceptable as long as the glowing area is recognized. In this way, the recognition of traffic lights in daylight is assessed by default, while at night it is more tolerant. To do so, we must first decide whether the detection center must be adjusted. The Area and the Shape measure can provide this information. Then, based on the structure of the lights (vertical or horizontal) and luminescence state marked in GT, the algorithm can adjust the position of the detection center in GT and calculate the distance similarity to correctly determine the detection performance. Examples of those calculations and data are in Figure 10 and Table 3.

**Figure 10.** Results for Traffic Light/Sign Recognition. Evaluation for traffic lights should focus only on the distance measure output in a situation when results of shape and area similarity are very low since there is no practical advantage of the whole case detection as long as the correct state of light is recognized.

**Table 3.** Rectangular similarity and Jaccard index for traffic signalization and traffic signs detected in Figure 10.

| TS | Jaccard | General | Distance | Area | Shape |
|----|---------|---------|----------|------|-------|
| 0 | 21.7% | 22.6% | 99.8% | 23.3% | 8.6% |
| 1 | 9.9% | 4.5% | 34.8% | 28.2% | 0.9% |
| 2 | 41.7% | 70.5% | 96.3% | 41.7% | 100.0% |
| 3 | 22.6% | 40.7% | 91.6% | 31.6% | 28.4% |
| 4 | 26.5% | 61.7% | 46.6% | 94.3% | 99.3% |
| 5 | 4.2% | 13.4% | 26.9% | 10.5% | 5.2% |
| 6 | 19.0% | 30.8% | 100.0% | 19.0% | 25.3% |
| 7 | 15.4% | 26.0% | 100.0% | 15.4% | 18.2% |

*5.4. Radar Data Evaluation*

The methodology presented by us can be adapted to work with results from perception modules and algorithms processing radar data. The results we obtain from these systems are again in the form of rectangles of a certain width and height, but this time represent a contour around the main vehicle seen from the bird's eye view. The radar results can be compared with those obtained from lidar, because they are considered much more precise and reliable—the lidar data play the role of GT. In addition, our methodology can be used to test whether the results of embedded system mounted in the cars overlap with the simulations, which is important for the development and testing of the driving control systems.

To compare the position, size, and shape of the objects, we can use the metrics described in the previous sections. But each object has assigned a velocity vector, which represents the direction and relative speed of the object. To assess the similarity between the velocity vectors for radar result $\vec{v}_r$ and lidar result $\vec{v}_l$, we can reuse the function used in the distance similarity measure (6). The argument for this function is the magnitude of the difference of vectors $d(\vec{v}_l, \vec{v}_r) = |\vec{v}_l - \vec{v}_r|$. Similarity $V$ of those vectors will be scaled based on the length of these vectors analogically as for the size of bounding boxes. Function *diag* is replaced by function that returns length of a vector for scale calculation.

$$V(\vec{v}_l, \vec{v}_r) = \exp\left[-\gamma d(\vec{v}_l, \vec{v}_r)^\delta\right]. \tag{20}$$

When the direction of both vectors is the same, the magnitude of the difference of the vectors is a simple difference between the numerical values of velocities of the objects. The same value can be obtained for many combinations of magnitudes and directions of vectors. But in all cases, the same length of the difference of vectors means that the compared objects will be in positions that are in the same distance from each other after a certain time has passed. Finally, we get a measure that is effective for comparing vectors, which directions and magnitudes take very different values.

## 6. Results of Measure Comparison

### 6.1. Feature Comparison

To compare our measure with selected known tools used to evaluate bounding boxes, we can check Table 4. Scale invariance provides a possibility to compare results for a different scale of compared objects. Sensitivity of position/size/shape/rotation means that deviation on this plane influences value of measure. Separation of these information allow us to know what type of deviation we are dealing with just from the analysis of results. Continuity for the separated boxes means that the measure is capable of providing information even for boxes that do not have common pixels. Adaptive focus is a feature that represents the ability to define certain parts of boxes as especially important in the process of evaluation. Utility of this feature is described deeply in Sections 5.2 and 5.3. Based on the provided features, the only measure that can compete with our novel proposal is the Jaccard Index. Despite of proofs from the previous chapters that our measure has a better appliance from automotive needs perspective, we will conduct an additional experiment that is an extended version of the measure quality analysis presented in [17,52].

**Table 4.** Comparison of features provided by quality measures for bounding box analysis.

| Measure Name | Scale Invariance | Position Sensitive/ Separable Influence | Size Sensitive/ Separable Influence | Shape Sensitive/ Separable Influence | Rotation Sensitive/ Separable Influence | Continuity for Separated Boxes | Adaptiv Focus |
|---|---|---|---|---|---|---|---|
| Hausdorff [53] | − | +/− | +/− | +/− | +/− | + | − |
| RobLoc [54] | + | +/+ | − | − | − | + | − |
| RobCor [54] | + | − | − | − | − | + | − |
| RobCom [54] | + | − | + | − | − | + | − |
| FOM [55] | + | +/− | +/− | +/− | +/− | + | − |
| Hafiane [52] | + | +/− | +/− | +/− | +/− | − | − |
| Jaccard (1) | + | +/− | +/− | +/− | +/− | + | − |
| Shape (3) | + | − | − | +/+ | − | + | − |
| Area (2) | + | − | +/+ | − | − | + | − |
| Distance (6) | + | +/+ | − | − | − | + | + |
| Velocity (20) | + | −/− | −/− | −/− | +/+ | + | + |
| GMOS (13) | + | +/+ | +/+ | +/+ | −/+ | + | + |

### 6.2. Artificial Interference Analysis

A Monte Carlo experiment was conducted to investigate the variability of the general similarity measure and the Jaccard index due to accidental deformations of the shape and position of the bounding box. In a series of experiments, we used more than 12,000 envelopes based on real data, derived from typical scenarios in the automotive industry. The main idea is to distort the shape and position of the box by combining continuous displacement of the box centroid with random displacement of corners of the envelope based on a two-dimensional normal distribution with different parameters. All displacements are in fact scaled by the diagonal length of a given bounding box to focus on influencing the final results of the distribution parameters. The results are presented in three examples of approaches.

#### 6.2.1. Position Shift

In Figure 11, we can see how the values of similarity measures behave when the shift of the box's center of gravity increases, as shown in Table 5. All corners of the box are shifted using a normal $N(0.05, 0.05)$ distribution. The initial value of the Jaccard measure starts at 0.6 for 0 shift of the center of gravity and drops when the shift increases without significantly changing the shape of the value histogram. The general measure for 0 shift value of the center of gravity creates a distribution centered at 0.9 and drops when the displacement increases with a significant change in the histogram shape of the value. The result is that

the general measure maintains high values for a wider range of displacement values—as desired, small shifts are not considered to be a significant change. For high displacement values, on the other hand, the general measure is much closer to 0 than the Jaccard index.

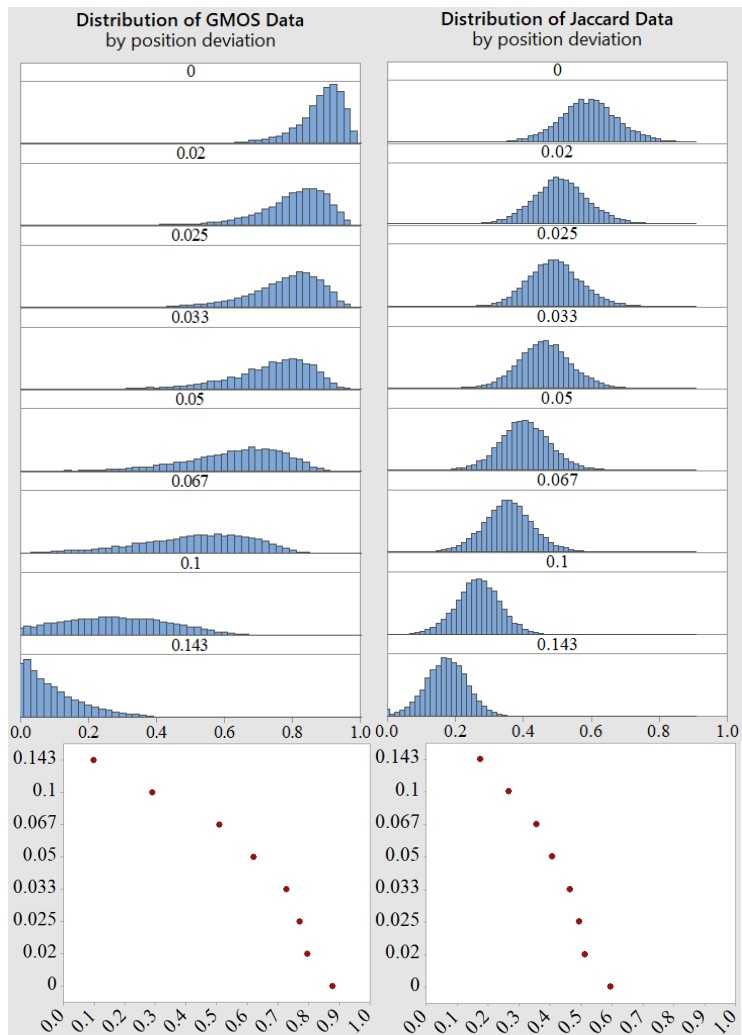

**Figure 11.** Comparison of frame data by different constant position shifts and corner disruption $N(0.05, 0.05)$. **Left side**—general measure histograms and mean comparison. **Right Side**—Jaccard index histograms and mean comparison.

**Table 5.** Values of varying parameters responsible for distortion.

| 0.000 | 0.020 | 0.025 | 0.033 | 0.050 | 0.067 | 0.100 | 0.143 |
|-------|-------|-------|-------|-------|-------|-------|-------|

### 6.2.2. Standard Deviation Influence

In Figure 12, we can see which measure is more sensitive to accidental shifting of the corner of the box. The standard deviation $\sigma$ of the corner shift was gradually increased as shown in Table 5. The center of the general measure histogram for shifts of the box's corner generated with normal distribution $\sigma = 0.020$, which we considered small, is close to 100% and drops as the corners are shifted more chaotically. Under the same conditions, the average value of the Jaccard index is practically constant and equals to 0.7 for $\sigma \in \{0, 0.020, 0.025, 0.033\}$.

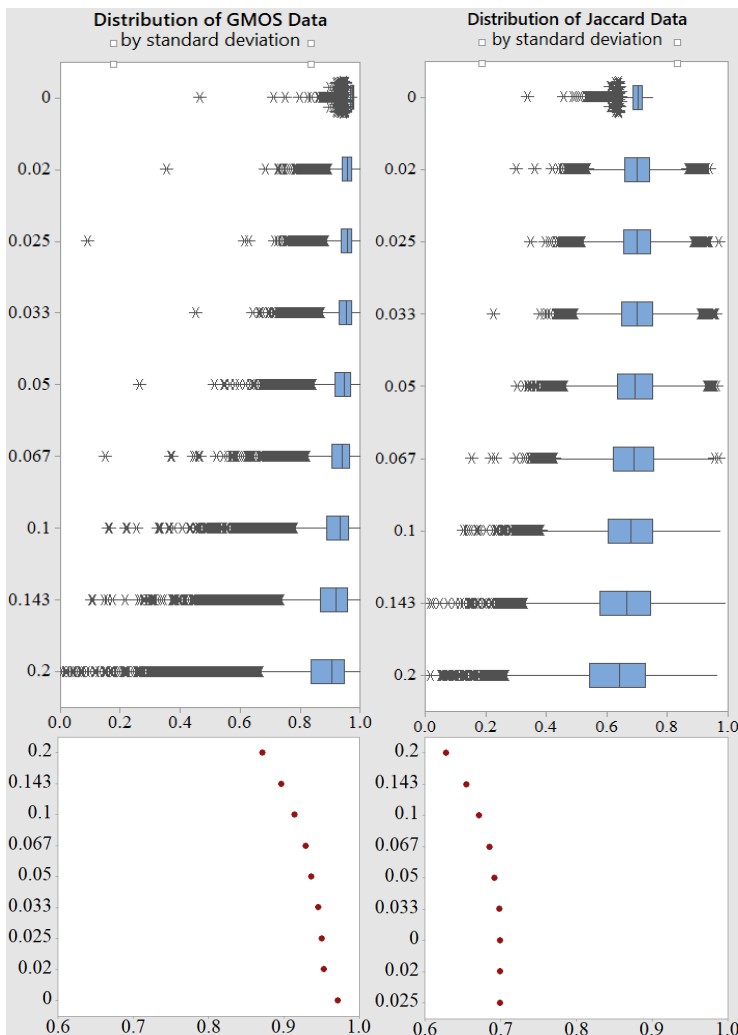

**Figure 12.** Comparison of frame data with position shift equal to 0.02 of diagonal length and corner disruption with mean equal to 0.02 and varying standard deviation. **Left side**—general measure histograms and mean comparison. **Right side**—Jaccard index histograms and mean comparison.

6.2.3. Position Shift in Event Summary

In Figure 13, we can see how the shifts of the centroid box from Table 5 affect the evaluation of events in the event tracking scenario. We used the standard approach described in the previous chapter and assumed that true positive detection occurs when the Jaccard index is greater than the threshold of 0.7. For these samples the corners of the boxes are shifted according to the normal distribution of $N(0.025, 0.025)$. For offset 0, the general measure values remain above 0.9 and decrease as the shift of the centroid increases. The Jaccard index is clearly misled by the shifting of field corners in combination with a threshold approach even for 0 shift of the center of gravity of the box. As the shift of the center of gravity increases, the value of the event's evaluation falls to 0.

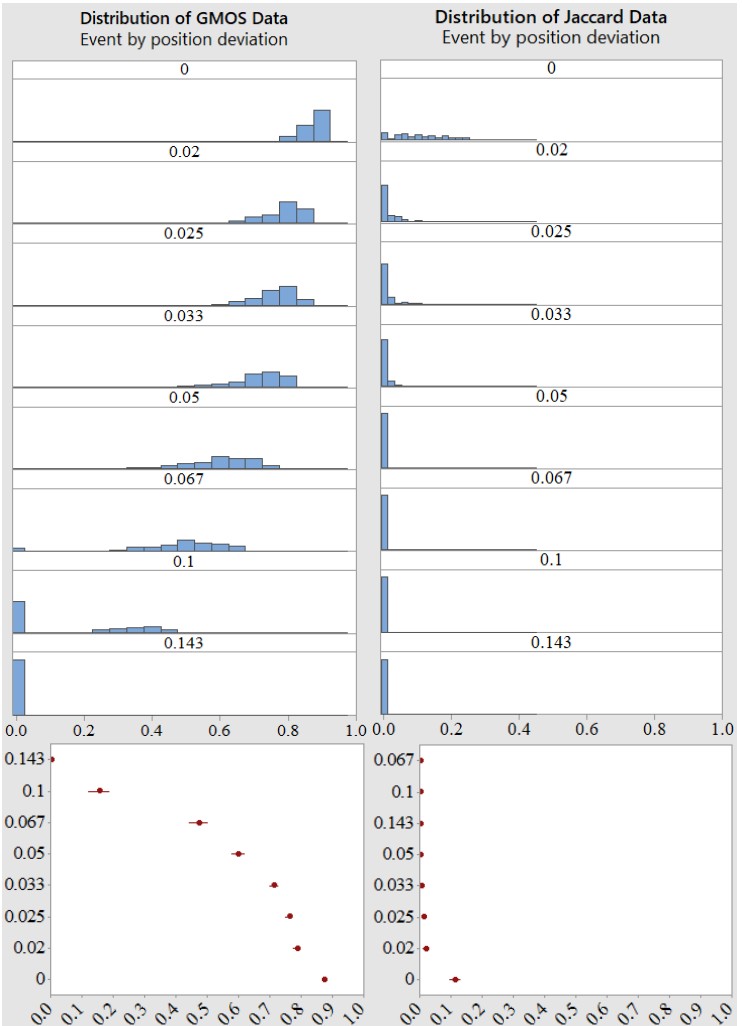

**Figure 13.** Comparison of event data by different constant position shift and corner disruption $N(0.025, 0.025)$. **Left side**—general measure histograms and mean comparison. **Right side**—Jaccard index histograms and mean comparison.

### 6.3. Statistical Difference

To check the difference between the Jaccard index results and our general measure results, we have conducted a permutation test (Portions of the information contained in this publication/book are printed with the permission of Minitab Inc. All such materials remains the exclusive property and copyright of Minitab Inc. All rights reserved). We chose 26 events detected in a single scene and evaluated them using the Jaccard index and the general measure, respectively. We drew 26 results from the uniform distribution, and calculated the difference in mean between the sample and both received groups of results. Then, we performed a permutation test for the differences in means with 10,000 repetitions. Both measures were significantly different from the random uniform sample. Then, we have calculated the difference in mean between the Jaccard index results and the general measure results (33.3%), and performed a permutation test with 10,000 repetitions Figure 14. The difference between those samples average values was significant ($p$ value < 0.0001). Additionally, we have conducted those permutation tests for both median and standard deviation. The difference between the standard deviation was not significant ($p$ value = 0.1371). On the other hand, medians of those samples was significantly different ($p$ value = 0.0027), which confirms a difference for position of the center of mass of data achieved by our created methodology. Additionally, the median analysis ensures that this conclusion is robust to potential outliers.

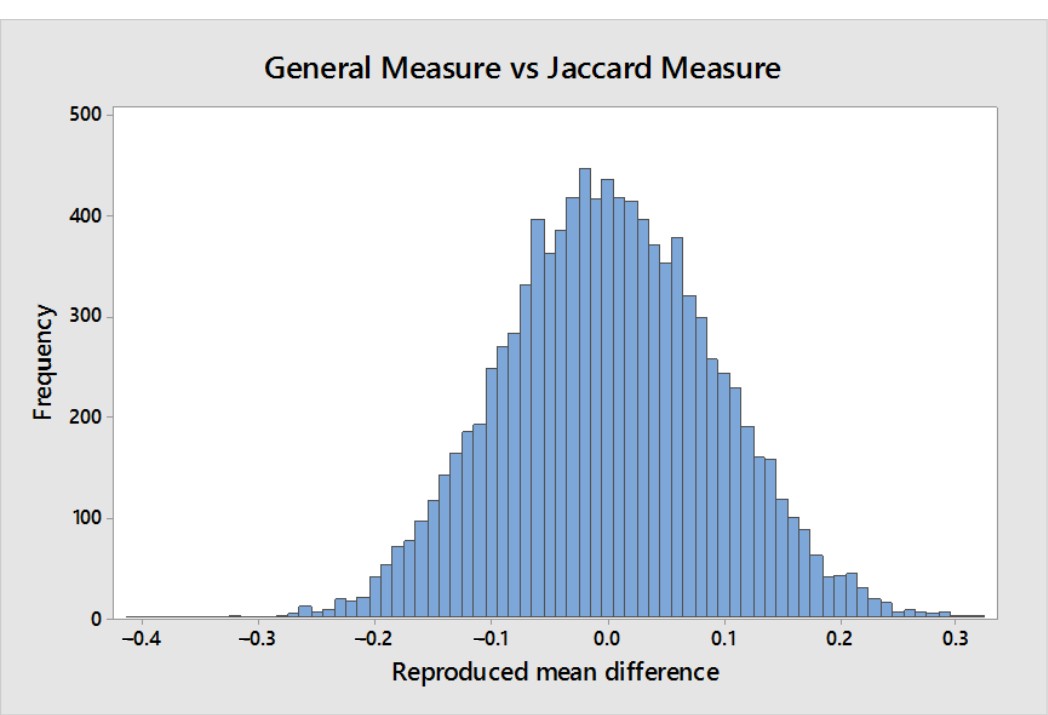

**Figure 14.** Permutation test result.

## 7. Conclusions

In this paper, we have proposed a new metric to evaluate the performance of vehicle detection systems based on front camera images mounted in car. Three almost orthogonal metrics have been defined to express similarity of area, shape, and distance. These metrics are combined into one synthetic general measure of similarity, but the individual use of each sub-metric is not excluded for certain tasks like matching data or to get more insights of comparison. Furthermore algorithm for assessing the quality of object described on timeline with sequence of bounding box based on analysis of first detection and tolerated delay was proposed. Experiments with real and artificial data have confirmed that our proposed local measure returns a result that is significantly different and more suited for car perception module development than that obtained using the Jaccard index. In addition, we have identified customizable options that can help automate solutions to various practical vehicle detection problems. The possible extension of the application to radar data was also indicated. We defended the thesis that using the methodology we provided, we can significantly reduce the time it takes to evaluate the detection software, not only by providing flexible automation options, but also by providing more detailed information about the detection quality at each level of the process. As a result, it can greatly limit number of situations where in order to confirm proper evaluation we must manually reach for raw data. This can significantly speed up data analysis and capture the weaknesses and limitations of the detection system under consideration.

**Author Contributions:** Conceptualization, P.K. and M.S.; methodology, P.K.; software, P.K.; validation, P.K. and J.I.; formal analysis, P.K.; investigation, P.K.; resources, P.K. and M.S.; data curation, P.K.; writing—original draft preparation, P.K.; writing—review and editing, P.K., J.I. and M.S.; visualization, P.K.; supervision, M.S. and J.I.; project administration, J.I.; funding acquisition, J.I. All authors have read and agreed to the published version of the manuscript.

**Funding:** Industrial PhD carried out at the Silesian University of Technology realized in cooperation with Aptiv Services Poland S.A.

**Data Availability Statement:** Not applicable.

**Conflicts of Interest:** The authors declare no conflict of interest.

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
