# Peer review of "Evaluation Methodology for Object Detection and Tracking in Bounding Box Based Perception Modules"

_electronics, doi:10.3390/electronics11081182_

Round 1

Reviewer 1 Report

Please find the comments referring to the paper as an attachment.

Reviewer 2 Report

The paper proposes three metrics to replace IoU. However, it is still hard to say that the proposes metrics are better. The proposes metric introduces many hyper-parameters, whose selection is not well discussed. For example, how are the coefficients in Eq.13 selected? It is not clear whether the metrics are robust the coefficients. Moreover, the hyper-parameters in like Eq.3 and Eq.9 should also be investigated.

The literature review is not complete. Some recent  tracking methods should be discussed. For example, Motion-attentive transition for zero-shot video object segmentation,Target-aware object discovery and association for unsupervised video multi-object segmentation.

Reviewer 3 Report

The pape proposed a metric to evaluate the performance of vehicle detection systems based on front camera images mounted in car. Three metrics have been defined to express similarity of area, shape, and distance.

The paper is well organized and well written.

Minor comment:

Section “Related Work” is too long and seems to have redundant contents. Such as Section 2.1. Preliminaries. It should be shorten.

Round 2

Reviewer 1 Report

All comments of the reviewer have been included in the revised version of the paper. I recommend publication this paper in its present form.

Reviewer 2 Report

The revision has addressed my concerns